# Broadly neutralizing plasma antibodies effective against autologous circulating viruses in infants with multivariant HIV-1 infection

Nitesh Mishra[1,7], Shaifali Sharma[1,7], Ayushman Dobhal [1,7], Sanjeev Kumar [1,4], Himanshi Chawla[1,5], Ravinder Singh[2], Muzamil Ashraf Makhdoomi[1,6], Bimal Kumar Das[2], Rakesh Lodha[3], Sushil Kumar Kabra[3] & Kalpana Luthra [1✉]

Broadly neutralizing antibodies (bnAbs) develop in a subset of HIV-1 infected individuals over 2–3 years of infection. Infected infants develop plasma bnAbs frequently and as early as 1-year post-infection suggesting factors governing bnAb induction in infants are distinct from adults. Understanding viral characteristics in infected infants with early bnAb responses will provide key information about antigenic triggers driving B cell maturation pathways towards induction of bnAbs. Herein, we evaluate the presence of plasma bnAbs in a cohort of 51 HIV-1 clade-C infected infants and identify viral factors associated with early bnAb responses. Plasma bnAbs targeting V2-apex on the env are predominant in infant elite and broad neutralizers. Circulating viral variants in infant elite neutralizers are susceptible to V2-apex bnAbs. In infant elite neutralizers, multivariant infection is associated with plasma bnAbs targeting diverse autologous viruses. Our data provides information supportive of polyvalent vaccination approaches capable of inducing V2-apex bnAbs against HIV-1.

[1] Department of Biochemistry, All India Institute of Medical Sciences, New Delhi 110029, India. [2] Department of Microbiology, All India Institute of Medical Sciences, New Delhi 110029, India. [3] Department of Pediatrics, All India Institute of Medical Sciences, New Delhi 110029, India. [4] Present address: ICGEB-Emory Vaccine Centre, International Centre for Genetic Engineering and Biotechnology, New Delhi, India. [5] Present address: Biological Sciences and the Institute for Life Sciences, University of Southampton, Southampton SO17 IBJ, UK. [6] Present address: Department of Biochemistry, Government College for Women, Cluster University Srinagar, Srinagar, India. [7] These authors contributed equally: Nitesh Mishra, Shaifali Sharma, Ayushman Dobhal. ✉email: kalpanaluthra@gmail.com

An effective human immunodeficiency virus-1 (HIV-1) vaccine that can curtail the AIDS pandemic is the need of the hour. The HIV-1 envelope glycoprotein (env), is a trimer of non-covalently linked heterodimers (gp120/gp41)₃, and is the primary target of broadly neutralizing antibodies (bnAbs). The bnAbs are capable of neutralizing diverse circulating variants of HIV-1 and are generated in rare subsets of infected individuals[1,2]. Passive administration of such bnAbs in animal models has shown protection from HIV-1 infection[3–5]. Recent studies conducted in HIV-1 infected individuals have shown that passive administration of bnAbs is effective in suppression of viremia[6–10]. HIV-1 bnAbs are categorized based on their recognition of five distinct and largely conserved epitopes on the envelope spike that are promising vaccine targets: the N160 glycan located within the V2 loop at the trimer apex (V2-apex), high mannose patch centered around N332 in the V3 region, the CD4 binding site (CD4bs), the membrane-proximal external region (MPER), and the N-glycans located at the gp120–gp41 interface[1,2]. A prolonged exposure to the viral env during natural infection has been implicated as a prerequisite for the development of bnAbs capable of neutralizing diverse viral strains, as is observed in select HIV-1-infected adults, who develop such bnAbs after a minimum of 2 to 3 years of infection[11–14].

In HIV-1 infected children, plasma bnAbs arise earlier in infection, and show higher potency and breadth compared to adults[15–19]. We observed the presence of cross-neutralizing antibodies in HIV-1 clade C chronically infected children[19] and recently generated a bnAb AIIMS-P01 from an elite pediatric neutralizer AIIMS_330[20]. Further, in this cohort of chronically infected children, AIIMS_329 and AIIMS_330, a pair of identical twins, showed elite plasma neutralizing activity[21]. A longitudinal analysis of the plasma antibody response and circulating viral strains showed the presence of diverse circulating viruses in both twins, with varied susceptibility to neutralization by plasma antibodies and bnAbs, irrespective of their similar genetic makeup and source of infection. Studies undertaken in infants have, however, documented that HIV-1 infected infants develop potent plasma bnAbs as early as one-year post-infection[17,18] suggesting that an effective vaccine in infants may perhaps be able to trigger the immune system and elicit an early bnAb response thus providing an impetus to evaluate the antibody response in a cohort of perinatally infected infants. Moreover, the bnAbs isolated from infected children show features atypical of adult bnAbs suggesting that the factors governing bnAb induction in infants are distinct from those in adults[20,22].

Infants infected via mother-to-child transmission (MTCT), with the well-defined genetic bottleneck leading to infection with a minor variant[23], provide a unique setting to understand the viral factors associated with induction of bnAbs. Herein, we evaluate the characteristic features of circulating viral strains in infants that show an early bnAb response to understand the antigenic triggers that drive B cell maturation pathways toward the induction of bnAbs.

## Results

### Identification of HIV-1 infected infant elite neutralizers.
In order to identify infants with potent plasma nAbs in our cohort of 51 infants, HIV-1 specific plasma bnAb breadth was assessed. We first evaluated nonspecific inhibitory effect by assessing the inhibition of MuLV infection in a TZM-bl based pseudovirus neutralization assay. Of the 51 plasma samples, 4 showed nonspecific inhibitory effect and were excluded from further analysis. To identify infants with early bnAb responses, we next performed plasma neutralization activity of the remaining 47 infants against a panel of 12 genetically divergent pseudoviruses[24,25],

representing global viral diversity, in order to capture plasma bnAbs targeting diversity encountered in the context of global HIV-1 pandemic. Cross-clade neutralization activity (CrNA), the ability to neutralize non-clade C pseudoviruses (different clade than the infecting clade) at ID₅₀ titers >50, was observed in 19 of the 47 infants at a median time of 12-months post-infection (p.i.) (range = 6–24 months) (Fig. 1a). Further, plasma neutralization activity against an 8-virus panel of Indian origin was assessed[21,25]. While the geometric mean titres were comparable for infants with both the global panel and Indian clade C panel, infants had higher breadth against Indian clade C panel (Supplementary Fig. 1a–c).

Though the plasma neutralization activity against the 12-virus global panel in infants was relatively broad with 21% (10/47) of infants neutralizing ≥50% of the pseudoviruses, their potency in comparison to plasma neutralization activity from previously characterized cohorts of chronically infected children and adults from our lab showed relatively lower magnitude (Fig. 1b, c), which prompted us to define pediatric elite neutralizers and broad neutralizers in the context of a modified breadth-potency matrix. For infant plasma samples, percent neutralization for each plasma-virus combination was recorded as a breadth-potency matrix at a fixed dilution of 1/50: >80% neutralization received a score of 3, >50% a score of 2, >20% a score of 1, and <20 received a score of 0. The maximum cumulative score for a given plasma was 36, and the neutralization score was given as the ratio of cumulative score for respective plasma to the maximum cumulative score, providing neutralization score on a continuous matrix of 0–1, with values closer to 1 implying strong plasma neutralization activity. The normalized neutralization scores, predictive of geometric mean titres and cross-clade neutralization, were calculated (Fig. 1d). A cut-off of 0.7 defined the 90th-percentile boundary and was used to define elite neutralizers (neutralizing ≥90% pseudoviruses), whereas a cut-off of 0.3 (75th-percentile) was used to define broad neutralizers (neutralizing ≥50% pseudoviruses) (Supplementary Fig. 2a, b). The neutralization scores for known elite and broad neutralizers from previously reported pediatric[19] and adult[26] cohorts were also calculated based on the same modified breadth-potency matrix, and the normalized neutralization score defined herein could categorize pediatric and adult elite and broad neutralizers (Supplementary Fig. 2b). Neutralization categories were defined based on normalized neutralization score with scores of ≥0.7 predictive of elite neutralization activity, scores in the range of 0.3–0.7 were predictive of broad neutralization activity, and scores in the range of 0.1–0.3 were predictive of cross-neutralization activity. Based on this scoring system, four infants were classified as elite neutralizers (AIIMS704, AIIMS706, AIIMS709, and AIIMS743) and six infants as broad neutralizers (AIIMS731, AIIMS719, AIIMS736, AIIMS744, AIIMS732, and AIIMS738).

### V2-apex targeting bnAbs predominated in infants.
Currently reported bnAbs primarily target five epitopes on env: glycan dependent sites in V2 and V3, CD4 binding site (CD4bs), gp120/gp41 interface, and MPER. In order to delineate the epitope specificities of the plasma bnAbs from these infant elite and broad neutralizers, we used the HIV-25710_2_43 mutant pseudoviruses containing key mutations within the epitope for V2-apex, V3-glycan, CD4bs, gp120–gp41 interface and MPER. The plasma bnAbs of majority of the infants (8/10) were directed against the V2-glycan, with two elite neutralizers (AIIMS704 and AIIMS706) showing multi-epitope dependency, a feature reported to be typically associated with chronic antigenic exposure[16,21] (Fig. 2a). For AIIMS709 and AIIMS736, no dependence on any of the five epitopes was observed. To further validate that the high frequency

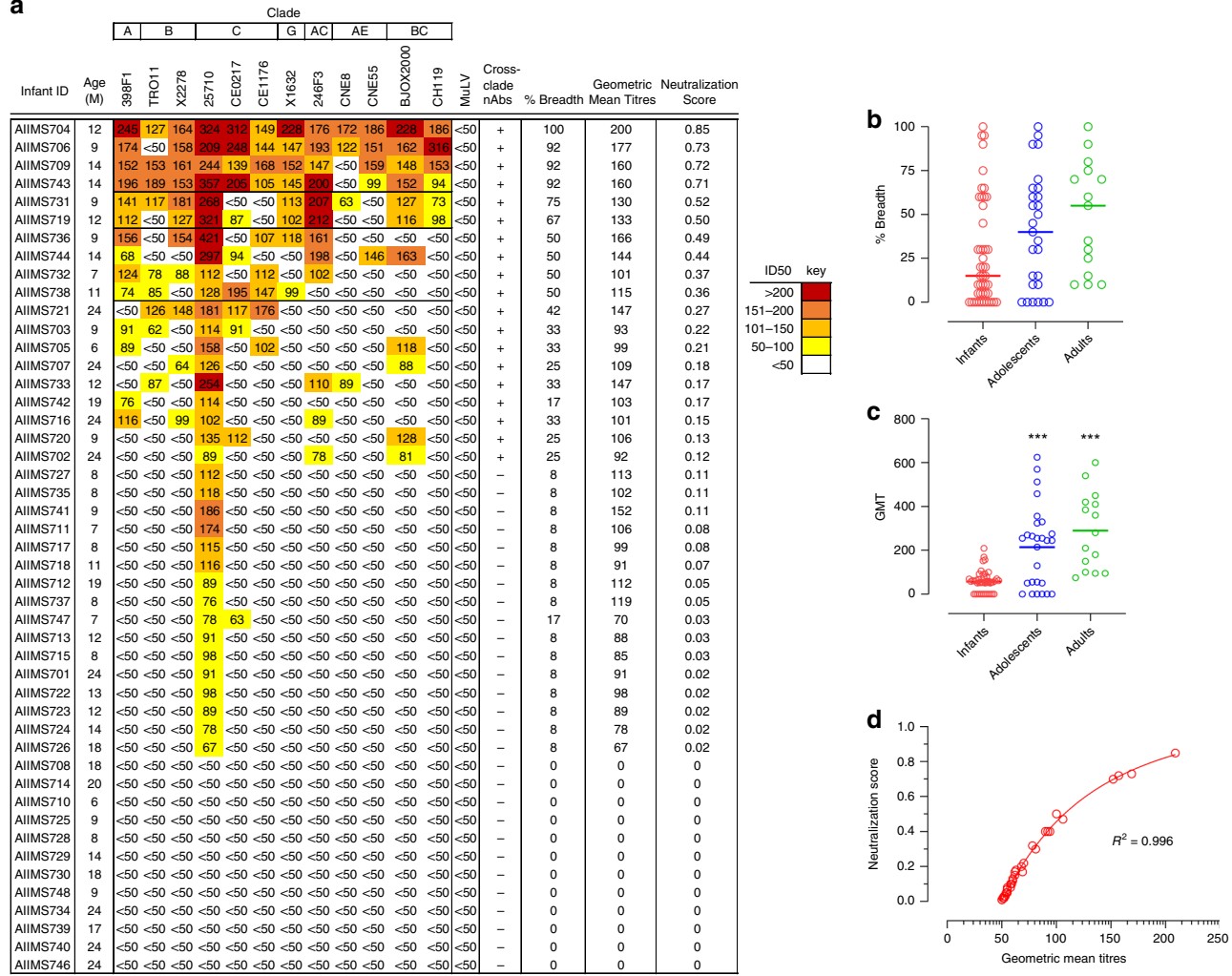

**Fig. 1 Identification of plasma bnAb-inducing infants. a** Heatmap representing HIV-1 specific neutralization titres (inverse plasma dilution) of plasma nAbs from 47 infant samples against the 12-virus global panel. $ID_{50}$ values are color-coded per the key given, with darker colors implying higher $ID_{50}$ titres. **b**, **c** Comparison of breadth (pseudoviruses showing >50% neutralization at 1/50 plasma dilution) and geometric mean titres of infants ($n = 47$) with previously established cohorts of chronically infected children (labeled 'adolescents') ($n = 27$) and adults ($n = 15$). Statistical difference between infant versus adolescents and adults was computed with two-tailed Mann–Whitney $U$ test. ***Represents a $p$-value less than 0.001. **d** Modified neutralization scores predict geometric mean titres.

of V2-apex targeting plasma nAbs were not a feature specific to HIV-25710_2_43 pseudoviruses, infant plasmas that showed V2-apex dependence were further mapped with 16055, CAP45 and BG505 N160A mutant pseudoviruses (Fig. 2b). For AIIMS706, additionally, N332A mutants of BG505, CAP256 and ConC were used as the plasma also had nAbs targeting V3-glycan (Fig. 2c). Though the extent of dependence varied from one pseudovirus to another, the expanded mapping showed similar trend as initial mapping done with HIV-25710_2_42, confirming high frequency of V2-apex plasma bnAbs in this cohort of infants. Interestingly, in AIIMS704, in addition to V2-apex targeting plasma nAbs, MPER dependence was also observed. MPER-directed bnAbs are rare in individuals with acute infection, highlighting the uniqueness of our observation of MPER plasma bnAbs in a 12-month old infant. To address whether the pan-neutralization of the global panel by AIIMS704 plasma nAbs were MPER mediated or an additive effect of having two distinct plasma nAb specificity, we depleted the MPER antibodies from AIIMS704, and checked the neutralization of global panel with MPER-depleted AIIMS704 plasma. Depletion of MPER antibodies from plasma was confirmed by binding ELISA against MPER-C peptide

(Supplementary Fig. 3a, b). MPER-depleted AIIMS704 plasma antibodies neutralized 50% of the global panel (6/12), and showed a modest 2.14-fold reduction in GMT titres across the global panel. All circulating recombinant viruses from the global panel (246F3, BJOX2000, CH119, CNE8, and CNE55) became resistant after depletion of MPER-specific plasma nAbs (Fig. 2d and Supplementary Fig. 3c). AIIMS704 MPER-depleted plasma nAbs were able to neutralize TRO.11, X2278, 25710, and CE1176. with $ID_{50}$ values comparable to that of undepleted plasma nAbs ($ID_{50}$ value of 112 vs 152 for X2278; $ID_{50}$ value of 188 vs 228 for 25710; $ID_{50}$ value of 78 vs 115 for CE1176 and $ID_{50}$ value of 122 vs 142 for TRO.11).

**Infant elite neutralizers had multivariant HIV-1 infection.** We next focused our analysis on three parameters (viral load, CD4 count, and duration of infection) to identify factors driving bnAb induction. No significant association was observed between CD4 counts or duration of infection with neutralization breadth (Supplementary Fig. 4a, b), suggesting that these parameters did not influence bnAb induction in this cohort of infants. High viral

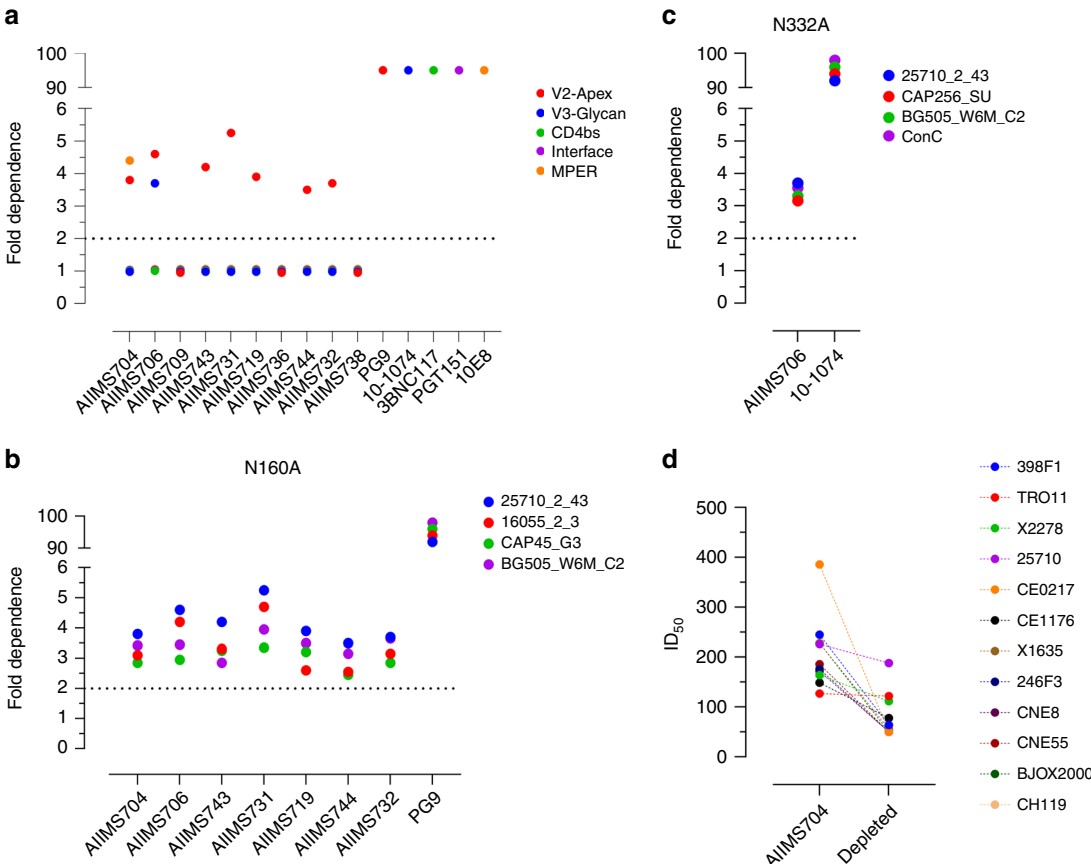

**Fig. 2 V2-apex targeting plasma nAbs predominate in HIV-1 infected infants. a** Epitope mapping done by mutant viruses in HIV-25710_2_43 backbone for V2-apex (N160A), V3-glycan (N332A), CD4bs (R456W), Interface (A512W-G514W) and MPER (W672L-F673L-T676A). PG9 (V2-apex), 10-1074 (V3-glycan), 3BNC117 (CD4bs), PGT151 (Interface) and 10E8 (MPER) bnAbs were used as positive controls. **b** V2-apex (N160A) dependence of infants with diverse pseudoviral backbones. PG9 (V2-apex targeting bnAb) was used as positive control. **c** V3-glycan (N332A) dependence of AIIMS706 plasma nAbs against diverse pseudoviral backbone. 10-1074 (V3-glycan targeting bnAb) was used as negative control. **d** Comparison of $ID_{50}$ titres of AIIMS704 plasma (undepleted) and MPER-peptide depleted plasma against the 12-virus global panel. Neutralization assays were performed in triplicates and repeated thrice. Average Ic50 values are shown and used for defining epitope dependence.

load showed negative correlation with neutralization breadth ($r = -0.497$, $p = 0.002$) (Supplementary Fig. 4c). Another factor influencing bnAb induction is the diversity in the viral envelope glycoprotein (env). Diversity in the viral env generated as a result of immune selection pressure (neutralization escape) or super-infection can independently drive bnAb induction[13,14,21]. To define viral diversity in the context of bnAb induction, we performed SGA analysis of env sequences[27] (V2C5 region of HIV-1 gp120, HXB2 position 6690–7757) from circulating viral variants in elite and broad neutralizers to assess the impact of viral diversity on bnAb induction. From elite and broad neutralizers, a total of 390 env gene sequences were obtained with more than or equal to 30 env sequences from each infant, giving a 90% confidence interval of sequencing circulating variants present at 5% population frequency (Table 1 and Supplementary Fig. 5a-j). In case of elite neutralizers, 45 SGA sequences were generated, further increasing the depth and confidence of sequencing to 95% at population frequency of 5%. Sequences containing large deletions or G-to-A hypermutations were excluded. All sequences were predicted to be clade C throughout the env gene reading frame, and had the highest phylogenetic relatedness to the reference sequence C.IN.95IN21068.AF067155 (GenBank accession number AF067155) complied in HIV database. Co-receptor usage was inferred based on V3 loop sequences, and all 390 sequences were predicted to use CCR5. Sequences were aligned, visually inspected using the highlighter tool, and maximum-likelihood phylogeny

tree were generated to identify the pattern of viral transmission. Pairwise raw distance distribution (nucleotide substitution per site) and env gene diversity (mean genetic distance) were assessed to calculate the intra-host diversity. All four elite neutralizers (AIIMS704, AIIMS706, AIIMS709 and AIIMS743) showed evidence of multivariant HIV-1 infection (distinct clusters on high-lighter plots and distinct branches on phylogeny tree with high degree of bootstrap support[28–33]), with one infant, AIIMS709, showing two highly divergent variants, plausibly due to super-infection, that needs to be further explored (Figs. 3a, b and 4a–d). SGA env sequences from broad neutralizers (AIIMS731, AIIMS719, AIIMS736, AIIMS744, AIIMS732, and AIIMS738) were monophyletic (Fig. 3a). Within-patient diversity in elite neutralizers ranged from 2.6 to 15.1 (Table 1). Such extent of within-patient diversity are typical of multivariant infections[27,29,30,32]. AIIMS709 had the maximum within-patient diversity, further confirming infection with highly divergent viruses. For broad neutralizers, within-patient diversity ranged from 0.3 to 0.7, a feature typical of infection with a single virus or multiple closely related viruses (Table 1). Significantly higher evolutionary divergence (nucleotide substitutions per site) was observed in elite neutralizers compared to broad neutralizers ($p = 0.0095$) (Supplementary Fig. 6a, b). Moreover, the multivariant infection in this infant cohort was significantly associated (odds ratio > 35, $p = 0.043$) with the development of plasma breadth, though a small sample size might have skewed the association.

**Table 1 HIV-1 env analysis of infant elite and broad neutralizers.**

| Infant_ID | Neutralization category | CD4 Count | Plasma Viral Load (Log) | SGA Amplicons (n) | Sex | Circulating variant | env gene diversity | env divergence |
|---|---|---|---|---|---|---|---|---|
| AIIMS704 | Elite | 2592 | 5.59 | 45 | M | 4 | 2.5 (0.6–3.2) | 0.026 |
| AIIMS706 | Elite | 1652 | 5.62 | 45 | M | 3 | 2.4 (0.4–3.6) | 0.029 |
| AIIMS709 | Elite | 2142 | 5.33 | 45 | M | 4 | 15.1 (0.4–16.6) | 0.097 |
| AIIMS743 | Elite | 1540 | 5.81 | 45 | M | 4 | 2.6 (0.4–3.5) | 0.022 |
| AIIMS731 | Broad | 1228 | 5.62 | 39 | M | 1 | 0.6 (0.1–1.5) | 0.002 |
| AIIMS719 | Broad | 1634 | 5.07 | 35 | F | 1 | 0.3 (0.1–0.7) | 0.004 |
| AIIMS736 | Broad | 2105 | 6.04 | 38 | F | 1 | 0.6 (0.6–1.2) | 0.002 |
| AIIMS744 | Broad | 2595 | 5.89 | 30 | M | 1 | 0.6 (0.2–1.1) | 0.002 |
| AIIMS732 | Broad | 1698 | 6.31 | 33 | M | 1 | 0.7 (0.6–1.3) | 0.003 |
| AIIMS738 | Broad | 1642 | 4.59 | 35 | M | 1 | 0.6 (0.3–0.77) | 0.003 |

*M*, male, *F*, female.
Circulating variants were estimated based on highlighter and bootstrapped maximum-likelihood trees (Figs. 3a, b and 4a–d). env divergence (average evolutionary divergence) among SGA amplicons for each infant was measured as the number of base substitutions per site from averaging over all sequence pairs within each group. env diversity (mean genetic distance) for each infant is shown as median with range, median (range).

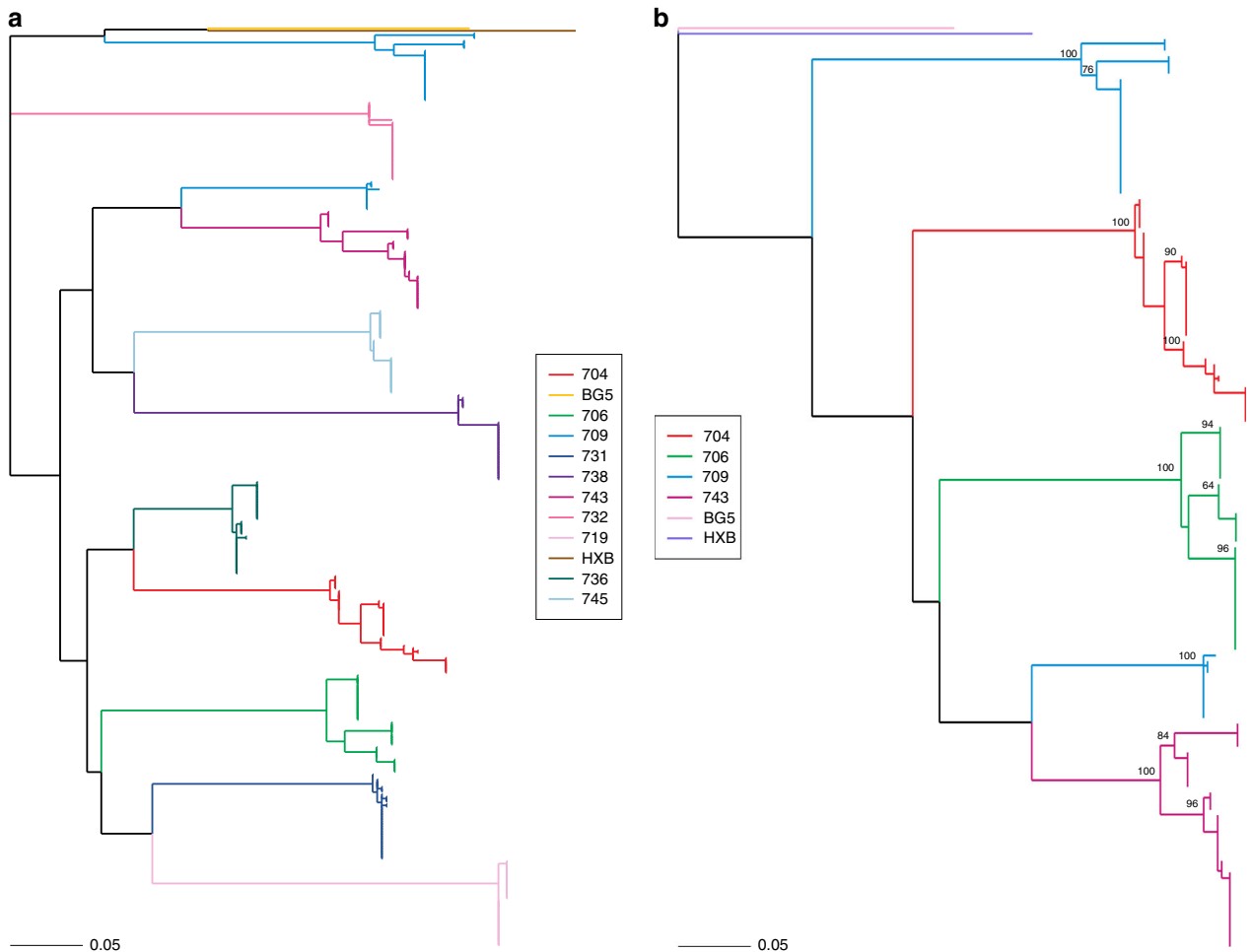

**Fig. 3 Multivariant infection in infants with elite plasma neutralizing activity. a** Maximum-likelihood tree of env SGA amplicons (V2C5 region, HXB2 position 6690–7757) of circulating viral variants from infants with elite and broad plasma neutralizing activity. BG505.W6M.C2 (Clade A, labeled BG5, yellow) and HXB2 (Clade B, labeled HXB, brown) were used as outgroups. Distinct multiple branches for AIIMS704 (red), AIIMS706 (green) AIIMS709 (blue), and AIIMS743 (deep pink) were observed. **b** Maximum-likelihood tree of full-length env sequences (HXB2 position 6225–8795) from the four elite neutralizers (AIIMS704, red; AIIMS706, green; AIIMS709, blue; and AIIMS743, deep pink). BG505.W6M.C2 (Clade A, labeled BG5, light pink) and HXB2 (Clade B, labeled HXB, purple) were used as outgroups. The numerals at the node represent bootstrap value. The horizontal scale bar represents genetic distance. nt, nucleotide.

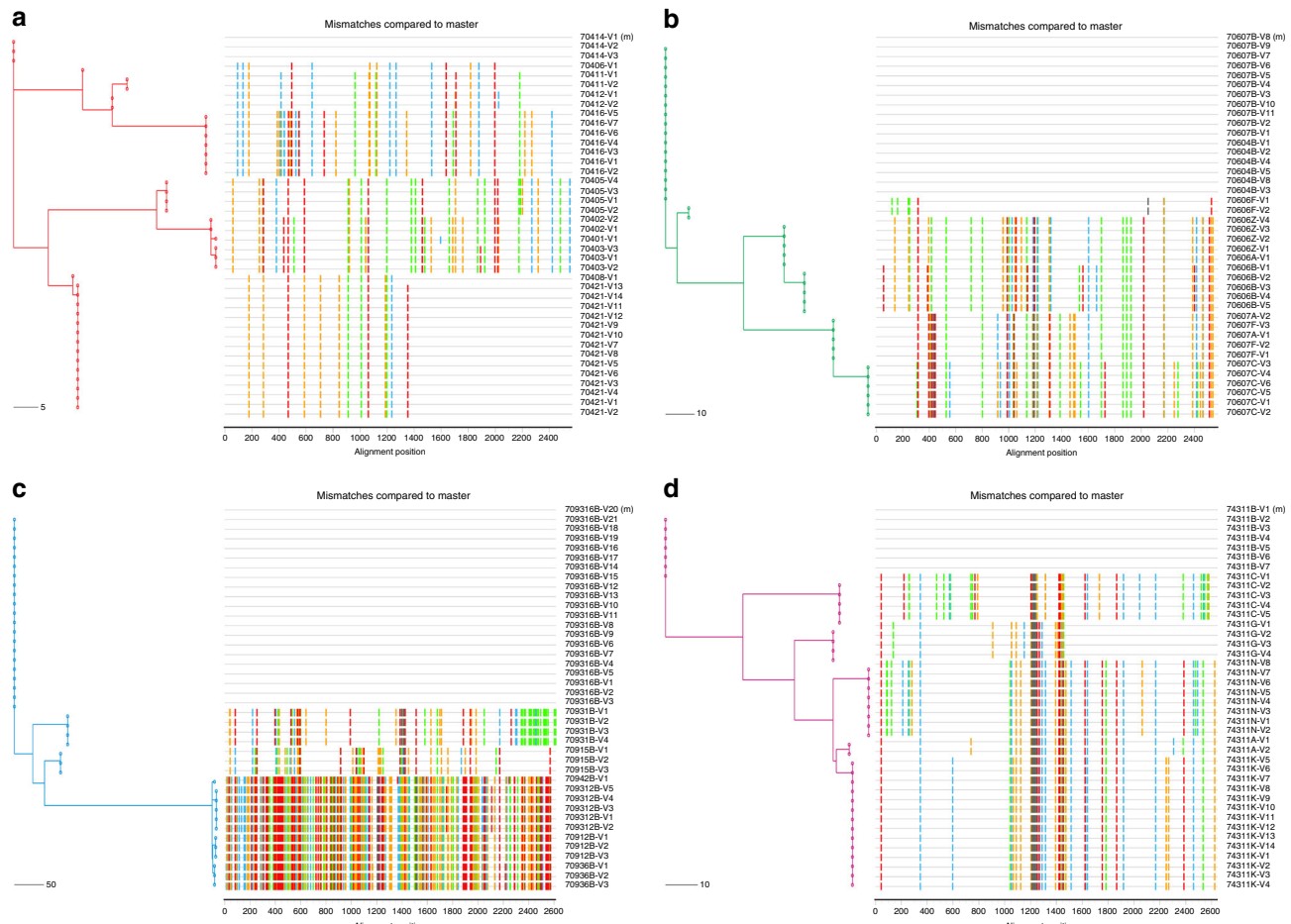

**Fig. 4 Distinct circulating viral variants in infant elite neutralizers. a–d** Highlighter plots with maximum-likelihood trees of 40 SGA env sequences from each infant suggests productive infection with more than two distinct viruses. Maximum-likelihood trees are color coded (AIIMS704—red, AIIMS706—green, AIIMS709—blue and AIIMS743—deep pink). Colored hash marks on each highlighter plot represent nucleotide difference (A—green; T—red, C—blue, and G—orange) compared to the sequence at the top of the plot. The horizontal scale bar represents genetic distance. nt, nucleotide.

**Contemporaneous viruses in infant elite neutralizers**. To gain insight and evaluate the impact of multivariant infection and decipher viral characteristics associated with the development of plasma nAbs in infant elite neutralizers, we next generated functional pseudoviruses from all four infant elite neutralizers. The generated pseudoviruses were tested for neutralization by autologous nAbs. Distinct viral populations in each of the infected infant showed varied susceptibility to autologous plasma nAbs, with several variants sensitive and others moderately sensitive to autologous plasma bnAbs (Fig. 5a), an observation we previously reported to be a plausible driver of elite plasma neutralization activity[21]. Moreover, none of the circulating viral strains in four infants were resistant to autologous plasma nAbs, suggesting env specific antibodies generated in context of two distinct viral variants can target epitopes on both envelopes. In addition, susceptibility of the pseudoviruses from these infant elite neutralizers to known bnAbs and non-nAbs was also assessed. Majority of the pseudoviruses showed similar neutralization susceptibility profile to V2-apex targeting bnAbs while the susceptibility to other bnAb classes varied between pseudoviruses. Notable sequence similarity was observed in the strand B and C of the V2-loop of the pseudoviruses from all four infant elite neutralizers (Supplementary Fig. 7). No reactivity was observed with non-nAbs for majority of the pseudoviruses, suggesting a well-ordered trimeric configuration (Fig. 5a). To study the antigenic and conformational characteristics of intact, native

env trimers, we transiently transfected HEK293T cells with respective env clones from each infant elite neutralizer that showed maximum susceptibility to autologous plasma nAbs and the panel of known bnAbs. No effect of sCD4 on the binding of 2G12, which binds an exposed glycan epitope on gp120, was observed suggesting CD4 binding did not induce gp120 dissociation (Fig. 5b–e and Supplementary Fig. 8). CD4-induced non-neutralizing antibodies, 17b and A32, that bind the open trimer showed no binding and in presence of sCD4, only weak binding could be observed. Predominant binding of trimer-specific bnAbs PGDM1400, and CAP256.25 was observed, though binding of 10-1074 and PGT151 did vary between envs. Thus, viral variants from these infant elite neutralizers were susceptible to bnAbs, showed diverse susceptibility profile to autologous plasma nAbs, and plausibly adopted closed trimeric conformations.

## Discussion

bnAbs responses in HIV-1 infected adults have been well established, whereas limited number of studies exist assessing these responses in HIV-1 infected children. Apart from select studies, information on the neutralization activity of plasma antibodies in infants is lacking. Herein, we observed cross-clade neutralizing activity in 42% of the infected infants, and even though we used a less stringent cut-off of 1/50, the virus panel utilized in this study

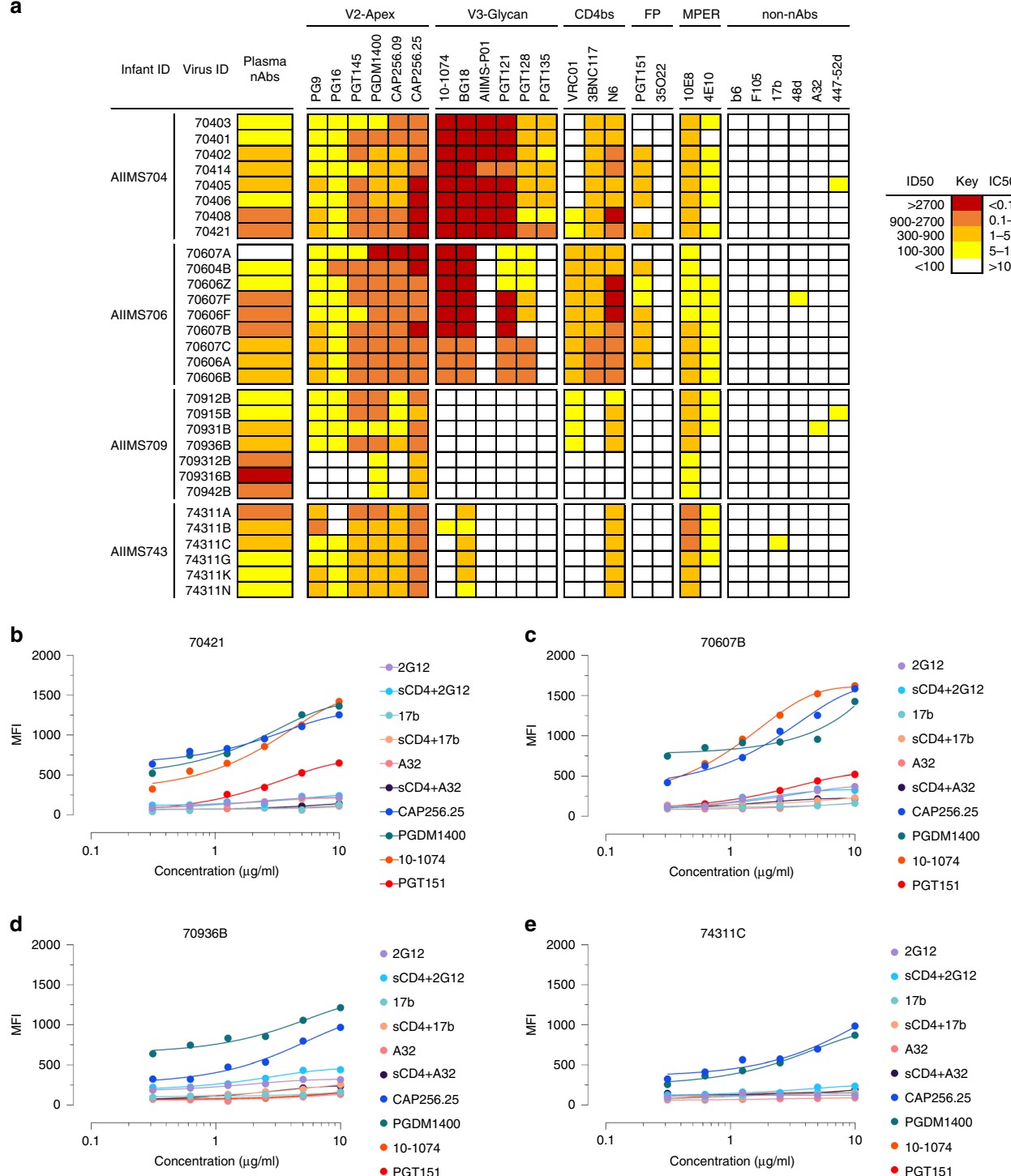

**Fig. 5 Neutralization profile and antigenic characteristics of pseudoviruses from infant elite neutralizers. a** Neutralization susceptibility of circulating viruses from elite neutralizers to autologous plasma nAbs and known bnAbs was assessed using TZM-bl cells. The potency of plasma and bnAbs is color coded per the key given. Most potent neutralization was seen with V2-apex targeting bnAbs. **b–e** Surface binding assay with varying concentration of trimer-specific V2-apex targeting bnAbs (PGDM1400, CAP256.25), V3-glycan targeting bnAb 10-1074, gp120–gp41 interface targeting bnAb PGT151, CD4-induced nnAbs (17b and A32, in presence and absence of sCD4), and gp120 outer domain targeting bnAb (2G12, in presence and absence of sCD4). Neutralization assays were performed in triplicates and repeated thrice. Average IC50 values were used for drawing heatmaps. All binding experiments were repeated thrice, and shown are the average MFI values. MFI, median fluorescence intensity.

had a relatively higher percentage of difficult to neutralize pseudo-viruses (normalized tier scores of 2.5–3)[24,34]. In the study conducted by Goo et al.[17], 71% of the infected infants showed cross-clade neutralization. Prevalence of cross-clade neutralization activity in

HIV-1 infected individuals from different cohorts has been shown to be 10 to 30%. The GMT values observed in this study were on the lower side, plausibly due to limited exposure to the antigen with a median infection duration of 12-months. Development of potent

plasma antibodies usually requires two to four years post-ser-oconversion, and is aided by chronic antigen exposure[11,12,19,26,35].

Currently reported bnAbs primarily target five epitopes on env: glycan dependent sites within the V2 and V3 (V2-apex and V3-glycan) regions, CD4 binding site (CD4bs), gp120/gp41 interface, and MPER[1,2]. The plasma antibodies of infant elite and broad neutralizers in this study were found to be directed against the V2-apex. Despite the high variability in terms of the sequence, glycosylation and length of the V2-apex of HIV-1 envelope, bnAbs directed against V2-apex are elicited relatively early and are one of the most potent classes of bnAbs[1,2,11,12]. The findings herein of a high frequency of V2-apex bnAbs in infants with cross-clade neutralization activity is in consensus with previous observations in infected children[16]. Given the relatively higher frequency, earlier induction, moderate level of somatic hyper-mutation, and consistent cross-clade neutralizing activity of V2-apex bnAbs in conjunction with cross-species conservation of the epitope given its critical function for trimer disassembly during viral entry[11,12,35–38], this bnAb epitope has been focus of recent immunogen design approaches. Our results further support the notion for exploring V2-apex targeting bnAbs, either as pro-phylactics or immunogen targeted induction, in the field of HIV-1 vaccinology. In two infant elite neutralizers, multi-epitope dependency was observed. Antibodies targeting multiple epitopes in children have been reported, though chronic exposure has been suggested as one of the mechanisms for the development of multiple antibody lineage[16,19]. Nevertheless, the exact factors that predispose children for the development of multiple bnAb lineages are unknown and need to be addressed. Antibodies against MPER are rarely elicited in children, plausibly due to the structural constraints in accessing the MPER as well as auto-reactivity of MPER bnAbs. Our findings of MPER nAbs in a 12-month old infant (AIIMS704) suggests similar bnAbs may be induced early with targeted vaccination strategies that may plausibly overcome the linked immune tolerance mechanisms blocking such nAbs.

Diversity in the viral env, either generated by immune selection pressure (neutralization escape) or superinfection, has been shown to be an independent driver of bnAb induction[13,14,21]. In our cohort, a significant association between multivariant infec-tion and elite plasma neutralization activity was seen, and though at the time of recruitment, all infants were in Fiebig stage VI, the extent of diversity observed could not be explained with estab-lished models of mutations gained due to escape mutations as a result of selection due to plasma nAbs[27,29,33]. Multivariant HIV-1 infection is more commonly seen in adults than in children who have acquired the infection by vertical transmission[27–29,39]. In both children and adults, the stringent genetic bottleneck for transmission often leads to infection by a single viral variant[40–42]. HIV-1 multivariant infection is defined as one person infected with two or more different HIV-1 strains. According to the timing of infection with the second strain, multivariant infections can be divided into co-infection[41] (acquisition of a second strain either simultaneously or before seroconversion) and superinfection[42,43] (acquisition of a second strain after ser-oconversion), but given the cross-sectional nature of this study, we could only categorically decipher the timeline of the multi-variant infections, and hence used the broad term of multivariant infection.

A better understanding of the mechanisms that determine the wide range of neutralization sensitivity and antigenic landscape of circulating primary HIV-1 isolates would provide important information about the natural structural and conformational diversity of HIV-1 env and how this affects the neutralization phenotype. Circulating viruses in infected individuals who develop potent plasma bnAbs characteristically develop resistance to autologous plasma nAbs as a result of the mutations acquired due to immune selection pressure. Herein, the circulating viruses sensitive to autologous plasma nAbs in all four elite neutralizers were observed. Interestingly, despite the presence of V2-apex targeting plasma nAbs, circulating viral variants of AIIMS704, AIIMS706 and AIIMS743 retained epitope-defining sequences and N-glycosylation sites at V2-apex. In addition, epitope defining key residues for all bnAb classes (V2-apex, V3-glycan, CD4bs, gp120–gp41 interface, and MPER) were retained on cir-culating viruses from AIIMS704 and AIIMS706. A fundamental challenge in HIV-1 vaccine strategy has been the development of native-like trimers capable of expressing bnAb epitopes while occluding immune-dominant non-neutralizing antibody epitopes. Preferential binding of bnAbs and not nnAbs has been correlated with efficient cleavage of env gp160 polypeptide into its con-stituent subunits (gp120 and gp41). For immunogen design, efficient cleavage of candidate envs is a desirable property. For majority of the envs from elite neutralizers, CD4i targeting antibodies (17b, 48d, and A32) showed negligible binding fol-lowing sCD4 triggering while trimer-specific bnAbs showed prominent binding, suggesting a stable and homogenous con-formational and antigenic state. In addition, recent line of evi-dence suggest that select HIV-1 viral variant have the capability to initiate bnAb responses[44]. Given that these infants generated remarkable antibody response within a year of infection, envs from these infants can be explored as potential immunogen candidates.

In this cross-sectional study on a small cohort of HIV-1 clade C infected infants, we have demonstrated an association of multivariant infection with the development of plasma bnAb response. Though exposure to two distinct viral variants in adults has been shown to be not sufficient to broaden neutralizing antibody responses[45,46], our findings of an early bnAb response in context of multivariant infection can be a distinct feature of infant immune response to HIV-1 infection. This needs to be validated in established cohorts of HIV-1 infected infants. Our data provide key evidence for exploring polyvalent vaccination approaches for pediatric HIV-1 infection. Polyvalent vaccines have been less explored due to immunodominance of HIV-1, which in turn can diminish the efficacy of vaccines[47,48]. We observed the plasma nAbs in infants with multivariant infection to target both variants, suggesting env specific antibodies gener-ated in context of two distinct viral variants can target epitopes on both envelopes. In addition, viral variants from these infants can be explored as candidate priming immunogens to elicit V2-apex targeting bnAbs. Furthermore, longitudinal analysis in established infant cohorts should be undertaken to understand how the immune system in infants responds to multiple HIV-1 strains, which will provide key insights for guiding the early development of such bnAbs. To conclude, our results further add to a growing body of literature suggesting infants may have different immu-nological tolerance mechanisms and may be permissive for the development of bnAbs.

## Methods

**Study design and participants**. The current study was designed to decipher the viral characteristics that influence the early induction of plasma bnAbs in HIV-1 infected infants. Antiretroviral naïve and asymptomatic HIV-1 infected infants below the age of 2-years visiting the Pediatric Chest Clinic, Department of Pediatrics, AIIMS during the duration of this study were recruited randomly. A total of 51 antiretroviral naïve and asymptomatic HIV-1 infected infants were recruited for this study. After written informed consent from guardians, blood was drawn in 3-ml EDTA vials, plasma was aliquoted for plasma neutralization assays, viral RNA isolation, and viral loads. The study was approved by institute ethics committee of All India Institute of Medical Sciences (IECPG-307/07.09.2017). The median age for infected infants was 12-months (IQR, 8–19), the median CD4 count was 1731 cells/mm$^{3-}$ (IQR, 1498–2562) and the median viral load on log scale was 5.804 RNA copies/ml (IQR, 5.331–6.301).

**Plasmids, viruses, monoclonal antibodies, and cells**. Plasmids encoding HIV-1 env genes representing different clades, monoclonal antibodies and TZM-bl cells were procured from NIH AIDS Reagent Program. 10-1074 and BG18 expression plasmids were kindly provided by Dr. Michel Nussenzweig, Rockefeller University, USA. CAP256.09, CAP256.25 and b6 were procured from IAVI Neutralizing Antibody Centre, USA. HEK293T cells were purchased from the American Type Culture Collection (ATCC).

**Neutralization assay**. Neutralization assays were carried out using TZM-bl cells, a genetically engineered HeLa cell line that constitutively expresses CD4, CCR5, and CXCR4, and contains luciferase and β-galactosidase gene under HIV-1 tat promoter. Neutralization studies included 47 heat-inactivated plasmas from infants, 19 bnAbs (PG9, PG16, PGT145, PGDM1400, CAP256.09, CAP256.25, 10-1074, BG18, AIIMS-P01, PGT121, PGT128, PGT135, VRC01, N6, 3BNC117, PGT151, 35O22, 10E8 and 4E10) and 6 non-nAbs (b6, F105, 17b, 48d, A32, 447-52D). Envelope pseudoviruses were incubated in presence of serially diluted heat-inactivated plasmas, bnAbs or non-nAbs for one hour. After incubation, freshly Trypsinized TZM-bl cells were added, with 25 µg/ml DEAE-Dextran. The plates were incubated for 48 h at 37 °C, cells were lysed in presence of Bright Glow reagent, and luminescence was measured. Using the luminescence of serially diluted bnAbs or plasma, a non-linear regression curve was generated and titres were calculated as the bnAb concentration, or reciprocal dilution of serum that showed 50% reduction in luminescence compared to untreated virus control. For plasma samples, percent neutralization for each plasma-virus combination was recorded as a breadth-potency matrix at a fixed dilution of 1/50: >80% neutralization received a score of 3, >50% a score of 2, >20% a score of 1, and <20 received a score of 0. Maximum cumulative score for a given plasma was 36, and neutralization score was given as the ratio of cumulative score for respective plasma to the maximum cumulative score, providing neutralization score on a continuous matrix of 0 to1, with values closer to 1 implying strong plasma neutralization activity. For epitope mapping, HIV-25710-2.43 pseudoviruses with key mutations within major bnAb epitopes were used[21], and greater than 3-fold reduction in $ID_{50}$ titres were classifies as dependence. HIV-25710-2.43 mutants included N160A (V2-apex), N332A (V3-glycan), R456W (CD4 binding site), A512W-G514W (Interface) and W672L-F673L (MPER). Extended mapping with N160A mutants of 16055_2_3, CAP45_G3 and BG505_W6M_C2, and N332A mutants of CAP256_SU, BG505_W6M_C2, and ConC was performed for samples that showed V2-apex and V3-glycan dependence respectively.

**Depletion of MPER plasma antibodies and binding ELISAs**. 4 wells in 96-well ELISA plates were coated with MPER-C peptide overnight at 4 °C. 100 µl of plasma was added and following a 45-min incubation, plasma was iteratively adsorbed in the remaining three wells. Briefly, 96-well ELISA plates (Corning, USA) was coated with 2 µg/ml of MPER-C peptides overnight at 4 °C. Coated plates were washed with PBS containing 0.05% Tween 20. Plates were blocked with 5% skimmed milk in blocking buffer. A 50-fold dilution of plasmas, was added, titrated in 2-fold dilution series, and incubated at 37 °C for 1 h. Unbound plasma antibodies were washed with wash buffer and plates were incubated with peroxidase conjugated goat anti-human IgG at a dilution of 1:1000. Following secondary antibody incubation, the wells were washed, and TMB substrate was added. After color development, reaction was stopped with 0.2 M $H_2SO_4$ and absorbance was measured at 450 nm.

**HIV-1 envelope sequences and phylogenetic analysis**. HIV-1 envelope genes were PCR amplified from plasma viral RNA by single genome amplification and directly sequenced commercially. Individual sequence fragments of SGA amplified amplicons were assembled using Sequencher 5.4 (Gene Code Corporation). Sub-typing for SGA sequences was performed with REGA HIV subtyping tool (400 bp sliding window with 200 bp steps size). Inter-clade recombination was examined with RIP 3.0 (Recombinant Identification Program) and with jpHMM. Nucleotide sequences were aligned with MUSCLE in MEGA X 10.1. Maximum-likelihood trees were computed with MEGA X 10.1 using a general-time reversal substitution model incorporating a discrete gamma distribution with five invariant sites. Evolutionary divergence within each infant's SGA sequence was conducted in MEGA X 10.1 and was calculated as number of base substitutions per site from averaging over all sequence pairs. Analyses were conducted using the Maximum Composite Likelihood model. The rate variation among sites was modeled with a gamma distribution (shape parameter = 5). This analysis involved 18 nucleotide sequences. Codon positions included were 1st + 2nd + 3rd+Noncoding. All ambiguous positions were removed for each sequence pair (pairwise deletion option). There were a total of 2622 positions in the final dataset.

**Nucleotide sequence accession numbers**. The SGA amplified HIV-1 envelope sequences used for inference of phylogeny and highlighter plots are available at GenBank with accession numbers MN703343–MN703404 and MT366192–MT366197.

**Generation of replication incompetent pseudoviruses**. Autologous replication incompetent envelope pseudoviruses were generated from AIIMS704, AIIMS706, AIIMS709 and AIIMS743 (elite neutralizers). Viral RNA was isolated from 140 µl of plasma using QIAamp Viral RNA Mini Kit, reverse transcribed, using gene

specific primer OFM19 (5′-GCACTCAAGGCAAGCTTTATTGAGGCTTA-3′) and Superscript III reverse transcriptase, into cDNA which was used in two-round nested PCR for amplification of envelope gene using High Fidelity Phusion DNA Polymerase (New England Biolabs). First round primers consisted of forward primer VIF2 (5′-GGGTTTATTACAGAGACAGCAGAG-3′) and reverse primer OFM19 (5′-GCACTCAAGGCAAGCTTTATTGAGGCTTA-3′). Second round primers consisted of forward primer ENVA (5′-CACCGGCTTAGGAATTTACT ATGGCAGGAAG-3′) and reverse primer ENVN (5′-TGCCAATCAGGGAAA AAGCCTTGTGTG-3′. The envelope amplicons were purified, and ligated into pcDNA3.1D/V5-His-TOPO vector (Invitrogen). Pseudoviruses were prepared by co-transfecting 1.25 µg of HIV-1 envelope containing plasmid with 2.5 µg of an envelope deficient HIV-1 backbone (PSG3Δenv) vector at a molar ratio of 1:2 using PEI-MAX as transfection reagent in HEK293T cells seeded in a 6-well culture plates. Culture supernatants containing pseudoviruses were harvested 48 h post-transfection, filtered through 0.4 µ filter, aliquoted and stored at −80 °C until further use. $TCID_{50}$ was determined by infecting TZM-bl cells with serially diluted pseudoviruses in presence of DEAE-Dextran, and lysing the cells 48 hours post-infection. Infectivity titres were determined by measuring luminescence activity in presence of Bright Glow reagent (Promega).

**Cell surface binding assay**. $1.25 × 10^5$ HEK293T cells seeded in a 12-well plate were transiently transfected with 1.25 µg of env-coding plasmids (pcDNA3.1 with cloned env/rev cassettes) using PEI-MAX. 48 h post-transfection, cells were harvested and per experimental requirement, distributed in 1.5 ml microcentrifuge tubes. For sCD4 triggering, 10 µg/ml of 2-domain sCD4 was added and incubated for 30 min at room temperature. For monoclonal antibody staining, 10 µg/ml of antibody was used and titrated 2-fold in staining buffer. 100 µl of primary antibody (HIV-1 specific monoclonals) were added to HEK293T cells expressing envs, and incubated for 30 min at room temperature. After washing, 100 µl of 1:500 diluted PE conjugated mouse anti-human IgG Fc was added, and after 30-min incubation, a total of 50,000 cells were acquired on BD LSRFortessa X20. Data was analyzed using FlowJo software (version v10.6.1).

**Statistics and reproducibility**. Two-tailed Mann–Whitney $U$ test and Kruskal–Wallis test were used for comparison of two and three parameters, respectively. All statistical analyses were performed on GraphPad Prism 8.3. A $p$-value of <0.05 was considered significant. Neutralization assays were performed in triplicates and repeated thrice. Average $ID_{50}$ values are shown and used for statistical comparisons. Binding ELISAs were performed in duplicates and repeated thrice. Average OD450 values were used for plotting curves. Surface binding assay was performed thrice and average PE-MFI (phycoerythrin-median fluorescence intensity) values were used for plotting curves.

**Reporting summary**. Further information on research design is available in the Nature Research Reporting Summary linked to this article.

## Data availability
The SGA amplified HIV-1 envelope sequences used for inference of phylogeny and highlighter plots are available at GenBank with accession numbers MN703343–MN703404 and MT366192–MT366197. All data required to state the conclusions in the paper are present in the paper and/or the Supplementary data. Source data are provided with this paper. Additional information related to the paper, if required, can be requested from the authors. Source data are provided with this paper.

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

## Acknowledgements
We thank all the study subjects for participating in this study. We are thankful to NIH AIDS Reagent program for providing HIV-1 envelope pseudovirus plasmids, bnAbs, non-nAbs and their expression plasmids, and TZM-bl cells, and Neutralizing Antibody Consortium (NAC), IAVI, USA for providing bnAbs. We are thankful to Dr. Michel Nussenzweig for providing 10-1074 and BG18 bnAb expression plasmids. This work was funded by Department of Biotechnology, India (BT/PR30120/MED/29/1339/2018). The Junior Research Fellowship (January 2016–December 2018) and Senior Research Fellowship (January 2019–October 2019) to N.M was supported by University Grants Commission (UGC), India.

## Author contributions
N.M. designed the study, performed SGA amplification, pseudovirus cloning, and neutralization assays, analyzed data, wrote the initial paper, revised, and finalized the paper. S.S., A.D., and M.A.M. contributed to SGA amplification, pseudovirus cloning, and neutralization assays. S.K. proposed the concept of neutralization scoring of infants. S.K. and H.C. expressed PGDM1400, CAP256.25, BG18, 10-1074 and AIIMS-P01 bnAb. R.S., and B.K.D. provided the immunological data of the HIV-1 infected infants. R.L. and Su.K.K. provided the samples of HIV-1 infected infants and provided patient care and management. S.S., A.D., S.K., H.C., and M.A.M. edited and revised the paper. K.L. conceptualized and designed the study, edited, revised and finalized the paper.

## Competing interests
The authors declare no competing interests.
