## [Peer Review File · Nature Communications]

Reviewers' Comments:

Reviewer #1:

Remarks to the Author:

This is a paper describing a cohort of HIV-1-infected infants and young children who developed broadly neutralizing antibodies (bnAbs) and the characterization of their characteristics. This contributes to a growing body of literature that suggests infants may have different immunological tolerance thresholds and may be permissive for the development of bnAbs. However, I cannot support this paper in the present version as there is both overreach in the wording implying a strength of evidence that is not present and a lack of detail about many of the analyses that make interpretation of the data impossible for the reviewer. The data may be correct and the findings important, but as presented I cannot judge the strength of the case being made.

1. Page 2 line 53: the statement that the data suggest that "similar bnAbs can be induced early with targeted vaccination" is not supported by the data shown. The data support that similar bnAbs *may* be elicited, but to say *can* would require a vaccination study.

2. Figure 2C is overinterpreted. The text (p3 lines 59-60) states that "High viral load emerged as a negative driver of neutralization breadth". High viral loads at the time of measurement were inversely correlated with breadth, but to say they were a driver would require more information than is presented, in particular, longitudinal data to show that viral loads were stable at the values shown.

3. Figures 2DEF are confusing. First, cross-neutralizers is not defined in the text; cross-neutralization is described as correlating with GMT and other parameters, but I do not see where it is defined except in Fig S1 where it is shown as +/- . Second, a case is made for a difference between male and female cross-neutralizers, but Fig 2D and Fig 2E both show overlapping distributions, while Fig 2F makes no sense—how are the values calculated and why are the dots different size? Third, the results of a statistical test are given in the text, but it is not explained to which figure this refers or what test was applied.

4. While I agree that the branching pattern shown in Fig S3 convincingly shows two circulating strains for Infant 709, the branching patterns for 704 and 706 are less convincing. With the absence of longitudinal data or deeper sampling, the patterns seen could easily be the result of undersampling. The data are suggestive, not definitive. This makes the claim of a strong association with multiple infection strains less convincing as well. This might be supported by a statistical analysis, but none is provided, and the interpretation of branching patterns appears to be visual inspection which could result in unintentional bias. Without all of the other data being presented in a manner similar to Fig S3, the reviewer is without data upon which to rest their judgement. On a side note, changing the colors of the branches between Figures 3 and S3 makes it hard to find the same infant in the two figures. In addition, in Fig 3, it is not clear where the data for CIN, 8BC, and 7BC come from as they do not appear in the tables or methods.

M. Anthony Moody

Reviewer #2:

Remarks to the Author:

The manuscript by Mishra, Sharma, Dobhal and colleagues builds on an emerging body of interesting data regarding the accelerated development of broadly neutralizing antibodies in pediatric HIV-infected donors. Here the authors assess a pediatric cohort for broadly neutralizing antibodies, and perform some limited mapping of specificities and viral sequence analysis.

Although there are some interesting observations in this paper, the analyses are generally fairly superficial and require additional work in order to substantially add to the HIV vaccine field.

Line 27 - the authors have used to global panel (as many others do) but also supplemented this with seven local Indian viruses. This results in quite a considerable over-estimate of breadth in this cohort compared to similar studies, as these plasma appear to preferentially neutralize these Indian viruses (Fig 1). I do agree that the breadth in the four elite neutralizers is remarkable for that age and warrants a much more detailed investigation of these four donors.

Line 28 - related to this, the authors claim that 19/47 (42% of donors) have cross-clade activity but this is often a consequence of neutralization of a single or two global viruses with low titer. This overstates the breadth of many of these donors.

Line 40 - Fig 1E - it's unclear how this analysis "confirms the validity of the panel and scoring system"

Line 42 - the mapping of specificities is weak, and makes use of a single envelope backbone only. The high levels of V2 antibodies reported here may well be a viral attribute of this single env clone, HIV-25710-2.43 (not well characterized in the literature). To be useful, the mapping needs to be significantly expanded. For example, as the authors mention, MPER antibodies are rare in children - is the breadth of donor AIIMS704 generally MPER-mediated or is that only true of HIV-25710-2.43 neutralization?

Line 59 - What do the authors mean by "antigenic saturation" in their cohort?

Also - Muenchoff, STM, 2016 (a key description of pediatric broadly neutralizing antibodies not cited here) report a correlation between VL and broadly neutralizing antibodies - what could explain these discrepant results? Is this a consequence of donor age, with children <5 years old experiencing a decline in VL before this stabilizes?

Line 70 - the authors state that they generated 30 SGA sequences per donor - why are only 4-10 sequences per donor shown on the tree? Are other measures of viral diversity derived (Fig 3 B) from the full set or smaller datasets?

Line 73 and title of paper - the authors argue that elite neutralizers are enriched for multivariant transmission. However, these sequences are not from the point of transmission and so multivariant transmission cannot be inferred. While AIIMS704 and 706 definitely show high levels of variability at this timepoint, this may well be a consequence of divergent escape pathways as has been shown for many donors with broadly neutralizing antibodies. Bootstraps should be added to the tree as a measure of statistical support.

In addition, on lines 80-85 the authors (incorrectly in my view) define multivariant infection as infection by "two or more different strains". With the exception of AIIMS709, this is not supported by Fig 3 where all other donors show phylogenetic linkage.

Line 77 - "with long branch lengths suggestive of highly diverse infecting viruses". This statement makes little sense. Please clarify.

Line 89 – this is true for adults too.

The section on phenotypic characterization of autologous clones is interesting, however no attempt is made to link the mapping with viral escape patterns. Furthermore, the description of autologous clones would benefit from a matched analysis of viral sequences.

Line 100 and elsewhere – it is unclear to me why the authors focus on V2 germline targeting. The only mapping performed here is for one clone – whether V2 antibodies truly mediate breadth in this cohort has not been shown. Furthermore, as the authors themselves point out, this would require a detailed understanding of whether these envelopes have the capacity to trigger V2 precursors.

Response to reviewers

We thank the reviewers for their valuable and pertinent suggestions that helped us immensely in revising our manuscript. Listed below are the responses to reviewers' comments where reviewer's comments are highlighted in bold while the answers are in normal font. The changes in the text are colored blue.

Reviewer #1 (Remarks to the Author):

This is a paper describing a cohort of HIV-1-infected infants and young children who developed broadly neutralizing antibodies (bnAbs) and the characterization of their characteristics. This contributes to a growing body of literature that suggests infants may have different immunological tolerance thresholds and may be permissive for the development of bnAbs. However, I cannot support this paper in the present version as there is both overreach in the wording implying a strength of evidence that is not present and a lack of detail about many of the analyses that make interpretation of the data impossible for the reviewer. The data may be correct and the findings important, but as presented I cannot judge the strength of the case being made.

1. Page 2 line 53: the statement that the data suggest that “similar bnAbs can be induced early with targeted vaccination” is not supported by the data shown. The data support that similar bnAbs *may* be elicited, but to say *can* would require a vaccination study.

The initial version of the manuscript was written as a brief report. The current version of the manuscript has been written as a full article and we have tried to ensure the usage of 'may' whenever we have discussed about the implications for vaccination.

2. Figure 2C is overinterpreted. The text (p3 lines 59-60) states that “High viral load emerged as a negative driver of neutralization breadth”. High viral loads at the time of measurement were inversely correlated with breadth, but to say they were a driver would require more information than is presented, in particular, longitudinal data to show that viral loads were stable at the values shown.

In the current version of the manuscript, the earlier statement, “high viral load emerged as a negative driver of the neutralization breadth,” has been replaced with “high viral load showed negative correlation with neutralization breadth (r = -0.497, p = 0.002)” (page 4, line number 138 – 139). In addition, figure 2a – c from the initial brief report have now been moved to supplementary figure 4a – c (page 23) in this revised manuscript.

3. Figures 2DEF are confusing. First, cross-neutralizers is not defined in the text; cross-neutralization is described as correlating with GMT and other parameters, but I do not see where it is defined except in Fig S1 where it is shown as +/-. Second, a case is made for a difference between male and female cross-neutralizers, but Fig 2D and Fig 2E both show overlapping distributions, while Fig 2F makes no sense—how are the values calculated and why

are the dots different size? Third, the results of a statistical test are given in the text, but it is not explained to which figure this refers or what test was applied.

In the revised full article version of the manuscript, cross-clade neutralization activity is defined in the text at first instant (page 2, line number 74 – 76) and the text reads, “Cross-clade neutralization activity (CrNA), the ability to neutralize non-clade C pseudoviruses (different clade than the infecting clade) at ID50 titers >50, was observed in 19 of the 47 infants at a median time of 12-months post-infection (p.i.) (range = 6 – 24 months) (Fig. 1a).”

Gender based comparison among the cross neutralizers is now omitted. (In the previous brief report, figure 2d represented the entire cohort (regardless of their neutralization potential) while figure 2e represented a comparison between male and female cross-neutralizers (19 of the initial 51 infants). Figure 2f represented the ratio of male or female cross neutralizers to the total number of male or female infants in the cohort).

4. While I agree that the branching pattern shown in Fig S3 convincingly shows two circulating strains for Infant 709, the branching patterns for 704 and 706 are less convincing. With the absence of longitudinal data or deeper sampling, the patterns seen could easily be the result of under sampling. The data are suggestive, not definitive. This makes the claim of a strong association with multiple infection strains less convincing as well. This might be supported by a statistical analysis, but none is provided, and the interpretation of branching patterns appears to be visual inspection which could result in unintentional bias. Without all of the other data being presented in a manner similar to Fig S3, the reviewer is without data upon which to rest their judgement. On a side note, changing the colors of the branches between Figures 3 and S3 makes it hard to find the same infant in the two figures. In addition, in Fig 3, it is not clear where the data for CIN, 8BC, and 7BC come from as they do not appear in the tables or methods.

In the revised manuscript, we increased the depth of sampling for all infant elite and broad neutralizers with 45 SGA sequences from elite neutralizers and ≥ 35 from broad neutralizers and is described in the text as, “From elite and broad neutralizers, a total of 390 env gene sequences were obtained with more than or equal to 30 env sequences from each infant, giving a 90% confidence interval of sequencing circulating variants present at 5% population frequency (Table 1 and Supplementary Fig. 5a – j). In case of elite neutralizers, 45 SGA sequences were generated, further increasing the depth and confidence of sequencing to 95% at population frequency of 5%. Sequences containing large deletions or G-to-A hypermutations were excluded.” (page 4, line 145 – 149).

The sequencing information is detailed in table 1 and supplementary figure 5 that summarizes the depth of SGA sequencing and generated a combined figure based on phylogenetic (bootstrap support of 1000 replicates) and highlighter plots (figure 4).

We followed the criteria as defined earlier in several reports (reference 20 – 25 in this manuscript) for identifying plausible multivariant infection; based on distinct clusters with high bootstrap support on phylogenetic trees (reference 20 -25), well defined clusters on highlighter plots and pairwise raw and

mean genetic distance between clusters of sequences of viruses derived from each infected individual (table 1, supplementary figure 6) (reference 19, 21, 22 and 24).

In the previous brief report version of the manuscript, we had inadvertently missed description of the full form for CIN (Indian Clade C) 8BC (CRF 08_BC) and 7BC (CRF 07_BC) and mention that reference sequences were compiled from HIV LANL database. In the current revised manuscript, we have excluded reference sequences to draw the phylogeny tree and instead, due to increased depth and increased number of SGA env sequences from infants, have made a phylogeny based only on infant sequences (though HXB2 and BG505_W6M_C2 sequences have been used to set out groups) (Figure 3a – b). Figure 3a shows a compiled phylogeny tree for all infant env sequences (elite and broad) while figure 3b shows a bootstrapped tree analysis for elite neutralizers (with increased depth of SGA env sequences).

Figure 4, with well distinguished clusters on the highlighter plot, is further suggestive of the presence of multivariant HIV-1 infection in the infant elite neutralizers (line 154 – 161).

M. Anthony Moody

Reviewer #2 (Remarks to the Author):

The manuscript by Mishra, Sharma, Dobhal and colleagues builds on an emerging body of interesting data regarding the accelerated development of broadly neutralizing antibodies in pediatric HIV-infected donors. Here the authors assess a pediatric cohort for broadly neutralizing antibodies, and perform some limited mapping of specificities and viral sequence analysis.

Although there are some interesting observations in this paper, the analyses are generally fairly superficial and require additional work in order to substantially add to the HIV vaccine field.

Line 27 - the authors have used to global panel (as many others do) but also supplemented this with seven local Indian viruses. This results in quite a considerable over-estimate of breadth in this cohort compared to similar studies, as these plasmas appear to preferentially neutralize these Indian viruses (Fig 1). I do agree that the breadth in the four elite neutralizers is remarkable for that age and warrants a much more detailed investigation of these four donors.

In the revised manuscript, we have segregated the neutralization data against the global panel (figure 1a) and Indian clade C panel (supplementary figure 1a) (page 2, line 70 – 74 and 76 – 77 respectively), so as to avoid overestimate of breadth, as stated by the reviewer.

The text now reads, “To identify infants with early bnAb responses, we next performed plasma neutralization activity of the remaining 47 infants against a panel of 12 genetically divergent pseudoviruses 16,17, representing global viral diversity, in order to capture plasma bnAbs targeting diversity encountered in the context of global HIV-1 pandemic. Cross-clade neutralization activity (CrNA), the ability to neutralize non-clade C pseudoviruses (different clade than the infecting clade) at ID50 titers >50, was observed in 19 of the 47 infants at a median time of 12-months post-infection (p.i.) (range = 6 – 24 months) (Fig. 1a). Further, plasma neutralization activity against an 8-virus panel of Indian origin was assessed 13,17. While the geometric mean titres were comparable for infants with both the global panel and Indian clade C panel, infants had higher breadth against Indian clade C panel (Supplementary Fig. 1a – c).”

In addition, we made a comparison of the neutralization breadth and potency against these two panels (supplementary figure 1b – c). The neutralizing activity against the Indian virus panel did tend to overshoot the breadth (although not significant), though no significant difference in the potency (GMT) was seen against either of the panels (page 2, line 77 – 79).

Line 28 – related to this, the authors claim that 19/47 (42% of donors) have cross-clade activity but this is often a consequence of neutralization of a single or two global viruses with low titre. This overstates the breadth of many of these donors.

We do agree that the inference of cross-clade activity observed in this cohort was also inclusive of infants with weak neutralization of few viruses; however taking into consideration the relatively high

tier scores (tier 2/3, normalized tier scores >2.5) of the global panel viruses and immaturity of infant immune responses, we have mentioned that 19/47 donors have cross-clade reactivity (page 2, line 74 – 76), and further discussed this as a limitation (line 202 – 210).

The revised text reads, “Herein, we observed cross-clade neutralizing activity in 42% of the infected infants, and even though we used a less stringent cut-off of 1/50, the virus panel utilized in this study had a relatively higher percentage of difficult to neutralize pseudoviruses (normalized tier scores of 2.5 to 3) 16,26. In the study conducted by Goo et. al. 9, 71% of the infected infants showed cross-clade neutralization. Prevalence of crNA in HIV-1 infected individuals from different cohorts has been shown to be 10 to 30%. The GMT values observed in this study were on the lower side, plausibly due to limited exposure to the antigen with a median infection duration of 12-months. Development of potent plasma antibodies usually requires two to four years post-seroconversion, and is aided by chronic antigen exposure.”

Line 40 – Fig 1E – it’s unclear how this analysis “confirms the validity of the panel and scoring system”

The line in the earlier brief report “The neutralization scores for known ENs and BNs from previously reported pediatric and adult cohorts were calculated based on the above criteria, confirming the validity of the panel and the scoring system defined herein to decipher broad and elite neutralizers,” (Page 2, line 39 – 41) has been explained in the full revised manuscript as “The neutralization scores for known elite and broad neutralizers from previously reported pediatric and adult cohorts were also calculated based on the same modified breadth-potency matrix and the normalized neutralization score defined herein could categorize pediatric and adult elite and broad neutralizers.”(line 95 – 98).

Based on established scoring methods in literature to define elite neutralizers, we have defined pediatric and elite neutralizers in earlier cohorts. In this manuscript, we used the current scoring method to retrospectively calculate neutralization scores based on the earlier data, and could successfully define elite and broad neutralizers.

Line 42 – the mapping of specificities is weak, and makes use of a single envelope backbone only. The high levels of V2 antibodies reported here may well be a viral attribute of this single env clone, HIV-25710-2.43 (not well characterized in the literature). To be useful, the mapping needs to be significantly expanded. For example, as the authors mention, MPER antibodies are rare in children – is the breadth of donor AIMS704 generally MPER-mediated or is that only true of HIV-25710-2.43 neutralization?

As suggested by the reviewer, in the revised manuscript, we have extended the mapping of epitope specificities with four distinct viral backbones (25710, 16055, CAP45 and BG505 for N160 dependence and 25710, BG505, CAP256, ConC for N332 dependence) (figure 2a – c) (page no 3 , line 114 – 119). Additionally, for AIMS704, we performed an MPER peptide depletion assay followed by neutralization assay against the global panel with the MPER depleted plasma and the results are detailed in the text as, “To address whether the pan-neutralization of the global panel by AIMS704 plasma nAbs were MPER mediated or an additive effect of having two distinct plasma nAb specificity, we de-

pleted the MPER antibodies from AIIMS704, and checked the neutralization of global panel with MPER-depleted AIIMS704 plasma. Depletion of MPER antibodies from plasma was confirmed by binding ELISA against MPER-C peptide (Supplementary Fig. 3a – b). MPER-depleted AIIMS704 plasma antibodies neutralized 50% of the global panel (6/12), and showed a 2.14-fold reduction in GMT titres. All circulating recombinant viruses from the global panel (246F3, BJOX2000, CH119, CNE8 and CNE55) became resistant after depletion of MPER-specific plasma nAbs (Fig. 2d, and Supplementary Fig. 3c).” (line 125 – 133).

Line 59 – What do the authors mean by “antigenic saturation” in their cohort?

Also - Muenchoff, STM, 2016 (a key description of pediatric broadly neutralizing antibodies not cited here) report a correlation between VL and broadly neutralizing antibodies – what could explain these discrepant results? Is this a consequence of donor age, with children <5 years old experiencing a decline in VL before this stabilizes?

As suggested by reviewer #1 that our statement would be valid if we had longitudinal data, we have omitted in the revised manuscript, the statement (line 59 in the brief report) concluding that high viral load negatively influenced neutralization breadth. We have cited the report by Muenchoff et. al. (reference 7) as suggested, in support for the line “In HIV-1 infected children, plasma bnAbs arise earlier in infection, and show higher potency and breadth compared to adults.” (line 46 – 47).

Though we have omitted in the revised manuscript the conclusion made in the earlier version based on viral load, our published and unpublished observations (Makhdoomi et. al., J. Gen. Virol. 2017; Aggarwal H et. al., Front. Immunol. 2018; Mishra et. al., J. Virol. 2019) with a longitudinal cohort of pediatric donors do suggest that viral load in the children are initially higher and stabilize in later stages of infection.

Line 70 – the authors state that they generated 30 SGA sequences per donor – why are only 4-10 sequences per donor shown on the tree? Are other measures of viral diversity derived (Fig 3 B) from the full set or smaller datasets?

As suggested by the reviewer, we have included all SGA env sequences from infant elite and broad neutralizers (a total of 390 SGA env sequences) to draw the phylogeny trees (figure 3 and 4) as well as to calculate the viral diversity. In addition, as suggested by reviewer #1, we have increased the depth of our SGA sequencing (table 1 and supplementary figure 3) and have mentioned in the text as, “From elite and broad neutralizers, a total of 390 env gene sequences were obtained with more than or equal to 30 env sequences from each infant, giving a 90% confidence interval of sequencing circulating variants present at 5% population frequency (Table 1 and Supplementary Fig. 5a – j). In case of elite neutralizers, 45 SGA sequences were generated, further increasing the depth and confidence of sequencing to 95% at population frequency of 5%. Sequences containing large deletions or G-to-A hypermutations were excluded.”(page 4, line 145 – 150).

Line 73 and title of paper - the authors argue that elite neutralizers are enriched for multivariant transmission. However, these sequences are not from the point of transmission and so

multivariant transmission cannot be inferred. While AIMS704 and 706 definitely show high levels of variability at this timepoint, this may well be a consequence of divergent escape pathways as has been shown for many donors with broadly neutralizing antibodies. Bootstraps should be added to the tree as a measure of statistical support.

In addition, on lines 80-85 the authors (incorrectly in my view) define multivariant infection as infection by “two or more different strains”. With the exception of AIMS709, this is not supported by Fig 3 where all other donors show phylogenetic linkage.

To address the issue pointed out by reviewer #2 as well as reviewer #1 that current sequence analysis (based on viruses that were not isolated from point of transmission) does not support the notion of multivariant infection, we first increased our depth of SGA sequencing. Secondly, we have created a combined figure based on phylogenetic tree (bootstrap support of 1000 replicates) and highlighter plots (figure 4). We have followed the criteria reported in several reports (reference 20 – 25 in this manuscript) for defining multivariant infection based on distinct clusters with high bootstrap support on phylogenetic trees (reference 20 -25), well defined clusters on highlighter plots and pairwise raw and mean genetic distance between clusters in individual infected individual (reference 19, 21, 22 and 24).

While we do agree that escape from plasma bnAbs may have potentially led to highly divergent viruses, complete env gene sequence analysis (as evident on highlighter plots in figure 4) suggests that the observed divergence was spread out across the entire env gene for all elite neutralizers. As escape typically occurs (though exceptions exist) in defined hotspot regions (within the contact region of bnAb epitope), the extent of diversity observed in these infants most likely appears due to infection with highly divergent viruses (line 234 – 238). The text now reads, “In our cohort, a significant association between multivariant infection and elite plasma neutralization activity was seen, and though at the time of recruitment, all infants were in Fiebig stage VI, the extent of diversity observed could not be explained with established models of mutations gained due to escape mutations as a result of selection due to plasma nAbs.”

However, in view of the concern raised, the title of the manuscript has been changed to “Broadly neutralizing plasma antibodies effective against diverse autologous circulating viruses in infants with multivariant HIV-1 infection.”

Line 77 – “with long branch lengths suggestive of highly diverse infecting viruses”. This statement makes little sense. Please clarify.

The statement, “with long branch lengths suggestive of highly diverse infecting viruses,” is omitted in the revised manuscript.

Line 89 – this is true for adults too.

The statement is revised as “In both children and adults, the stringent genetic bottleneck for transmission often leads to infection by a single viral variant.” (line 240 – 241).

The section on phenotypic characterization of autologous clones is interesting, however no attempt is made to link the mapping with viral escape patterns. Furthermore, the description of autologous clones would benefit from a matched analysis of viral sequences.

While autologous viruses showed diverse susceptibility to bnAbs, except for a single virus (70607A, figure 5a) in the case of AIIMS706, all autologous viruses in infants remained sensitive to autologous plasma bnAbs (figure 5a) which greatly limited our ability to map viral escape patterns. Further, the hierarchy of neutralization sensitivity we observed for autologous plasma varied from 2 – 3-fold which we assumed was not a significant enough difference to pinpoint escape mechanisms, as any significant conclusion (based on changes in ID50 or IC50) requires a higher value of fold-change. Moreover, due to extensive diversity between autologous viruses, despite similar neutralization sensitivity, undertaking detailed mutagenesis studies was not feasible in this study.

In case of loss of sensitivity to known bnAbs, defined escape mutations (such as loss of N160 or N332 glycans, alteration of fusion peptide) were observed.

Line 100 and elsewhere – it is unclear to me why the authors focus on V2 germline targeting. The only mapping performed here is for one clone – whether V2 antibodies truly mediate breadth in this cohort has not been shown. Furthermore, as the authors themselves point out, this would require a detailed understanding of whether these envelopes have the capacity to trigger V2 precursors.

The section in the earlier brief report about the V2-apex germline targeting was added as all generated pseudoviruses were susceptible to V2-apex bnAbs while diverse susceptibility to other bnAb classes was observed. In the revised full article version, we have removed the section discussing the potential for V2-apex germline targeting.

Reviewers' Comments:

Reviewer #1:

Remarks to the Author:

This is a revised paper describing broadly neutralizing plasma antibodies and their characterization in a cohort of children from India. This revision is extensive enough that it is almost a new paper, but more importantly, this version is vastly improved. The authors have done a good job of addressing the concerns of prior reviewers. Furthermore, they have tempered some of the overreach of the prior version, and where they have retained strong conclusions, they have supported it with new data.

The manuscript needs some copyediting; there are about a dozen places where I saw some sentences that could be tightened. I only have one concern remaining that needs to be addressed.

Line 37: I'm not sure that the references cited support the statement that human clinical trials have demonstrated protection from infection with passively administered bnAbs. Certainly, animal studies have shown that, but the AMP trial results are not yet published.

Michael Anthony Moody

Reviewer #2:

Remarks to the Author:

The revised manuscript by Mishra et al has been revised to address both my concerns and those of a second reviewer, largely through the addition of mapping data and more sequencing data. The revised manuscript is improved with the sequence analysis improved, and many of the more tenuous interpretations either removed or tempered.

The revised analysis of the breadth of this cohort, removing the Indian viruses, does indeed highlight the more average levels of breadth in this cohort, better highlighting the more interesting elite neutralizers.

The authors have added more mapping data, as requested. Unfortunately, these experiments have not been able to define the true polyclonality of bNAb responses in these donors which would have required all plasma to be mapped with all mutants. Some of the data are hard to understand – for example in AIMS704, in Figs 2A and B, all viruses tested seem to be neutralized via V2, whereas in Fig 2D, all viruses are neutralized by MPER with the exception of TRO. How do the authors explain these discordant data? In addition, there are technical concerns – for Figure 2B the figure suggests an N160K mutants were used. The mutant is well known to confer massive enhancement of neutralization sensitivity to HIV positive plasma. In addition, were N332K mutants really used in Figure 2C? The text refers to N160A and N332A – this might make more sense. The absence of controls is a concern here.

The extension of the sequencing analysis is welcome addition. I would have preferred to see equal sampling between elite and broad neutralizers, but agree that the overall interpretation, that elite neutralizers show greater levels of diversity, is of interest. It seems that Figure 3A and B are showing different length sequences (V2C5 vs gp160) so the fact that the topologies of the trees are so similar is somewhat surprising. Also, despite very different branch lengths, there scale of the two trees is the same (0.05). Is this correct? There additional discrepancies between Figures 3 and 4 - In Figure 3, the smaller branch of AIMS709 shows variation, whereas the highlight plot indicates that all of these sequences are identical?

The value of Supp Fig 7 is unclear – is this all sequences from elite neutralizers combined?

Response to Reviewers

We profoundly thank the editor and reviewers for critically reviewing the manuscript and for their pertinent comments. Please find below the point wise responses and details of modifications made in the manuscript based on the suggestions/comments. Listed below are the responses to reviewers' comments where reviewer's comments are highlighted in bold while the responses are in normal font. The changes in the text are colored blue.

Reviewer #1 (Remarks to the Author)

This is a revised paper describing broadly neutralizing plasma antibodies and their characterization in a cohort of children from India. This revision is extensive enough that it is almost a new paper, but more importantly, this version is vastly improved. The authors have done a good job of addressing the concerns of prior reviewers. Furthermore, they have tempered some of the overreach of the prior version, and where they have retained strong conclusions, they have supported it with new data.

The manuscript needs some copyediting; there are about a dozen places where I saw some sentences that could be tightened. I only have one concern remaining that needs to be addressed.

Q. Line 37: I'm not sure that the references cited support the statement that human clinical trials have demonstrated protection from infection with passively administered bnAbs. Certainly, animal studies have shown that, but the AMP trial results are not yet published.

R. We have updated the references and reframed the line, '*Passive administration of such bnAbs in animal models and in recently concluded human clinical trials with bnAbs alone, or in combination with antiretroviral therapy, have shown protection from HIV-1 infection,*' and it now reads as '*Passive administration of such bnAbs in animal models has shown protection from HIV-1 infection* (reference Barouch 2013, PMID:24172905; Gautam et al., 2018, PMID:29662199; Calenda et al., 2019, PMID: 31083697). *Recent studies conducted in HIV-1 infected individuals have shown that passive administration of bnAbs is effective in suppression of viremia* (Caskey et al., 2015, PMID: 25855300; Caskey et al., 2017, PMID: 28092665; Bar-On et al., 2018, PMID: 30258217; Mendoza et al., 2018, PMID: 30258136; Cohen et al., 2019, PMID:31393868).' (page number 1, line numbers 35 – 37)

Reviewer #2 (Remarks to the Author)

The revised manuscript by Mishra et al has been revised to address both my concerns and those of a second reviewer, largely through the addition of mapping data and more sequencing data. The revised manuscript is improved with the sequence analysis improved, and many of the more tenuous interpretations either removed or tempered.

Q. The revised analysis of the breadth of this cohort, removing the Indian viruses, does indeed highlight the more average levels of breadth in this cohort, better highlighting the more interesting elite neutralizers. The authors have added more mapping data, as requested. Unfortunately, these experiments have not been able to define the true polyclonality of bNAb responses in these donors which would have required all plasma to be mapped with all mutants.

R. While we do agree that mapping the epitope specificity of plasma nAbs from all infants would have enabled us to precisely define the polyclonality of plasma nAbs, we were restricted due to the weak neutralization responses seen in the remaining 37 infants (infants that were not in the elite and broad neutralizer group). As is evident in supplementary figure 1a, only 4 of the remaining 37 infants neutralized the 25710 pseudovirus with ID50 titres >150; moreover, due to the limited volume of plasma available from infants, we could not perform neutralization assay below 1/50 dilution of plasma. Typically, a 3-fold reduction in ID50 titre with a mutant virus is used to define epitope specificity. Given the weak neutralizing activity of the 37 plasma samples, the precise epitope specificity of these samples could not be evaluated.

Neutralization fingerprint analysis (computational prediction using correlation matrix) (Georgiev et al., 2013, PMID: 23661761) did suggest that even infants with weak neutralization response targeted V2-apex (PG9-like antibodies). Herein we did not undertake this predictive analysis due to availability of neutralization data from only a limited panel of pseudoviruses. Neutralization fingerprinting requires neutralization data with large panel of pseudoviruses to accurately predict epitope specificity.

Q. Some of the data are hard to understand – for example in AIMS704, in Figs 2A and B, all viruses tested seem to be neutralized via V2, whereas in Fig 2D, all viruses are neutralized by MPER with the exception of TRO. How do the authors explain these discordant data?

R. For AIMS704, MPER depletion had minimal impact on neutralization of TRO.11. In figure 2d, ID50 values for X2278 (green dot), CE1176 (black dot) and 25710 (violet dot) are also comparable. In figure 2d, we had erroneously assigned the ID50 value for 25710 as 378, instead of the actual value of 228 (Supplementary figure 3c) and have rectified the same as suggested.

In addition, as evident from the ID50 values described in supplementary figure 3c, AIMS704 MPER depleted plasma nAbs were able to neutralize TRO.11, X2278, 25710 and CE1176. with ID50 values relatively comparable to that of undepleted plasma nAbs (ID50 value of 112 vs 152 for X2278; ID50 value of 188 vs 228 for 25710; ID50 value of 78 vs 115 for CE1176 and ID50 value of 122 vs 142 for TRO.11). MPER depletion had the most significant impact on neutralization of circulating recombinant

forms (246F3, BJOX2000, CH119, CNE8 and CNE55). Such degree of variation in ID50 is observed within assay replicates due to the non-linear nature of TZM-bl based neutralization assays (Montefiori 2009, PMID: 19020839).

The above observations were stated in the text as, “*MPER-depleted AIIIMS704 plasma antibodies neutralized 50% of the global panel (6/12), and showed a modest 2.14-fold reduction in GMT titres across the global panel. All circulating recombinant viruses from the global panel (246F3, BJOX2000, CH119, CNE8 and CNE55) became resistant after depletion of MPER-specific plasma nAbs (Fig. 2d, and Supplementary Fig. 3c).* (page number 4, line numbers 129 – 133).’

We have now appended the text accordingly, and it now reads as, ‘*AIIIMS704 MPER depleted plasma nAbs were able to neutralize TRO.11, X2278, 25710 and CE1176. with ID50 values comparable to that of undepleted plasma nAbs (ID50 value of 112 vs 152 for X2278; ID50 value of 188 vs 228 for 25710; ID50 value of 78 vs 115 for CE1176 and ID50 value of 122 vs 142 for TRO.11).* (page number 4, line numbers 133 – 136).’

Q. In addition, there are technical concerns – for Figure 2B the figure suggests an N160K mutants were used. The mutant is well known to confer massive enhancement of neutralization sensitivity to HIV positive plasma. In addition, were N332K mutants really used in Figure 2C? The text refers to N160A and N332A – this might make more sense. The absence of controls is a concern here.

R. We used N160A and N332A mutants for all pseudoviral backbones tested and had inadvertently labelled the figures with N160K and N332K. The same panel of control bnAbs as shown in figure 2a were also used for extended epitope mapping in figure 2b and 2c. We have now revised figure 2b and 2c and included the appropriate bnAb controls (PG9 for V2-apex and 10-1074 for V3-glycan).

Q. The extension of the sequencing analysis is welcome addition. I would have preferred to see equal sampling between elite and broad neutralizers, but agree that the overall interpretation, that elite neutralizers show greater levels of diversity, is of interest. It seems that Figure 3A and B are showing different length sequences (V2C5 vs gp160) so the fact that the topologies of the trees are so similar is somewhat surprising. Also, despite very different branch lengths, there scale of the two trees is the same (0.05). Is this correct?

R. Despite the different length sequences (V2C5 vs gp160), the fact that topologies of the trees are so similar suggest that they depict the same evolutionary history. Further, despite very different branch lengths, the scales of the two trees is the same (0.05), thereby depicting the branching history of common ancestry, with a similar pattern of branching (topology) irrespective of branch lengths.

Another plausible reason for similar topology can be attributed to the fact that C1 and the gp41 region in HIV-1 do not acquire significant number of mutations compared to variable regions, and since the sequences used for figure 3a contained all variable regions except V1 compared to figure 3b, it might have influences the similar topology of the trees in figure 3a and 3b.

The scale of the two trees is 0.05.

Q. There are additional discrepancies between Figures 3 and 4 - In Figure 3, the smaller branch of AIMS709 shows variation, whereas the highlight plot indicates that all of these sequences are identical?

R. For highlighter plots, a master sequence is used to compare differences across the sequence alignment. In case of AIMS709, two highly divergent viruses (roughly 21% difference in the env gene sequence) were observed. Since highlighter plots can only be drawn using master sequences, in case of AIMS709, this inadvertently led to generation of a plot where minor differences in the viral cluster (from which we did not generate the master sequence) would not lead to significant alteration in the highlighter alignment. There are minor differences in the highlighter plot for the smaller branch of AIMS709 (depicted with black arrows) but they are not visibly prominent due to the scale of figure 4.

For AIMS704, AIMS706 and AIMS743, these issues were not observed on the highlighter plot since these viral variants showed roughly 3 – 5% difference between viral clusters.

Q. The value of Supp Fig 7 is unclear – is this all sequences from elite neutralizers combined?

R. Supplementary figure 7 was generated to show that viral variants from all elite neutralizers had similar V2-apex bnAb epitope features, notably the presence of N160 glycans and lysine-rich strand B and C. We have appended the text accordingly.

‘Notable sequence similarity was observed in the strand B and C of the V2-loop of the pseudoviruses from all four infant elite neutralizers (Supplementary Fig. 7).’ (line numbers 186 - 187, page number 5).

The legend of the figure now reads, ‘Weblogo plot showing the amino acid frequency plots of V2-loop (HXB2 amino acids 156 – 177) from all SGA amplicons of four infant elite neutralizers. Key glycan (N156 and N160) as well as lysine-rich strand B and C were well-conserved.’

Reviewers' Comments:

Reviewer #2:

Remarks to the Author:

I am satisfied with the revised manuscript.